# Hydrogen Bonding (Base Pairing) in Antiviral Activity

**DOI:** 10.3390/v15051145

**Published:** 2023-05-10

**Authors:** Erik De Clercq

**Affiliations:** Rega Institute for Medical Research, KU Leuven, 3000 Leuven, Belgium; erik.declercq@kuleuven.be

**Keywords:** hydrogen bonding, base pairing, acyclic nucleoside phosphonates (ANPs), FV-100, sofosbuvir, remdesivir, lethal mutagenesis (error catastrophe)

## Abstract

Base pairing based on hydrogen bonding has, since its inception, been crucial in the antiviral activity of arabinosyladenine, 2′-deoxyuridines (i.e., IDU, TFT, BVDU), acyclic nucleoside analogues (i.e., acyclovir) and nucleoside reverse transcriptase inhibitors (NRTIs). Base pairing based on hydrogen bonding also plays a key role in the mechanism of action of various acyclic nucleoside phosphonates (ANPs) such as adefovir, tenofovir, cidofovir and O-DAPYs, thus explaining their activity against a wide array of DNA viruses (human hepatitis B virus (HBV), human immunodeficiency (HIV) and human herpes viruses (i.e., human cytomegalovirus)). Hydrogen bonding (base pairing) also seems to be involved in the inhibitory activity of Cf1743 (and its prodrug FV-100) against varicella-zoster virus (VZV) and in the activity of sofosbuvir against hepatitis C virus and that of remdesivir against SARS-CoV-2 (COVID-19). Hydrogen bonding (base pairing) may also explain the broad-spectrum antiviral effects of ribavirin and favipiravir. This may lead to lethal mutagenesis (error catastrophe), as has been demonstrated with molnutegravir in its activity against SARS-CoV-2.

## 1. Introduction

Hydrogen bonding resulting in the base pairing of adenine (A) with thymine (T) and of guanine (G) with cytosine (C) is essential for the formation of double-helical DNA. That this hydrogen bonding would play a decisive role in the mechanism of action of a broad variety of antiviral agents is the subject of the present review article.

The era of antibiotics started accidentally in 1928 with the antibacterial activity of a *Penicillium* product that was later identified as penicillin, an enzyme that is targeted at the glycopeptide transferase in the synthesis of the peptidoglycan of the bacterial cell wall. After penicillin, several other antibiotics were discovered that are targeted at various other crucial steps (i.e., protein and nucleic acid synthesis) in the life cycle of bacterial organisms.

The era of antivirals started in a totally different way. In fact, in 1959, the first antiviral drug ever described was originally identified as an antitumor agent, assumed to act as a thymidine analogue, thereby inhibiting cellular DNA synthesis. The compound concerned was idoxuridine (IDU). Later, IDU became the first antiviral agent drug to be licensed for the (topical) treatment of herpes simplex virus (HSV) infections.

Idoxuridine (IDU) was at the origin of a wide variety of nucleoside analogues with antiviral potential against virtually all viruses, whether belonging to the DNA virus or RNA virus genera. A common denominator in these nucleoside analogues is that they all achieve their antiviral activity through a unique propensity, i.e., they engage in base pairing (through hydrogen bonding) with a complementary heterocyclic ring (i.e., adenine with thymine or uracil and guanine with cytosine). This base pairing (through hydrogen bonding) is further documented in the present review article.

## 2. Hydrogen Bonding

Hydrogen bonding has been traditionally accepted as the underlying force in the formation of base pairs: adenine-thymine (A:T) and guanine-cytosine (G⋮C), generally referred to as Watson–Crick type of base pairing, requiring two or three hydrogen bonds, respectively. The AT and GC base pairing has been inherently linked to the identification of the double helix by Watson, Crick [1] and Wilkins; however, in double-helical DNA, the number of adenine residues equaled that of thymine, and the number of guanine residues equaled that of cytosine, which had been previously observed by Erwin Chargaff. Nevertheless, the hydrogen bonding between adenine (A) and thymine (T) and that between guanine (G) and cytosine (C) have been traditionally characterized as the Watson–Crick type of base pairing (Figure 1A,B). This designation has also been extended to double-stranded RNA, where the adenine-uracil base pairing has been substituted for adenine-thymine.

## 3. Arabinosyladenine (Ara-A, Vidarabine)

Ara-A was first synthesized by Lee et al. [2] and originally considered an antitumor agent [3]. That it was antivirally active against the herpes simplex virus (HSV) and vaccinia virus was first reported by Privat de Garilhe and De Rudder [4]. Ara-A became the first nucleoside analogue shown to be effective upon systemic administration in the therapy of VZV infections [5]. One year later, Ara-A therapy was also shown to be effective in the therapy of biopsy-proved herpes simplex encephalitis [6]. Ara-A (vidarabine) would later be the subject of a controlled trial comparing it with acyclovir in the treatment of neonatal HSV infection [7].

Following its phosphorylation to the 5′-mono-, 5′-di- and 5′-triphosphate, the latter (Ara-ATP) (Figure 2) can act as an inhibitor/substrate of the viral DNA polymerase. This implies that ara-A should be able to engage in base pairing (hydrogen bonding) with thymine (Figure 2).

From ara-A, ara-AMP was developed to increase its aqueous solubility, but this modification did not resolve the intrinsic problem associated with ara-A, which is its rapid deamination (by the ubiquitous adenosine deaminase) to arabinosylhypoxanthine (ara-Hx), which is antivirally inactive.

## 4. 5-Substituted 2′-Deoxyuridines (IDU, TFT, BVDU)

The 5-substituted 2′-deoxyuridines 5-iodo-2′-deoxyuridine (idoxuridine, IDU) and 5-trifluoromethyl-2′-deoxyuridine (trifluorothymidine, TFT) (Figure 3) were the first antiviral nucleoside analogues ever commercialized for clinical use, albeit topically in the treatment of herpetic keratitis. IDU was first synthesized by Prusoff [8] as a potential antitumor agent. It was assumed to inhibit DNA synthesis, the site of inhibition probably occurring at either the 5′-monophosphate or 5′-triphosphate level [9]. That IDU also had antiviral potential was first shown by Herrman [10], using the vaccinia virus and herpes simplex virus (HSV) as the virus challenges. One year later, Kaufman had already reported that IDU was effective (clinically) in the local (topical) treatment of HSV keratitis in rabbits and humans [11]. Two years later, Kaufman and Heidelberger announced that TFT was also effective in the topical treatment of HSV keratitis [12]. Following IDU and TFT, a wealth of other 5-substituted 2′-deoxyuridines and other nucleoside analogues were shown to be active against HSV replication [13], the most potent against HSV-1 being BVDU [(*E*)-5-(2-bromovinyl)-2′-deoxyuridine (BVDU)] [14]. That BVDU could be successfully administered to patients with varicella-zoster virus (VZV infection) was reported [15] before its anti-VZV activity in cell culture was fully documented [16].

IDU, TFT and the 5-substituted 2′-deoxyuridines at large are metabolized (phosphorylated) successively to the 5′-mono-, 5′-di- and 5′-triphosphate form–for BVDU, the first phosphorylation step is carried out by the HSV-encoded thymidine kinase (TK), whereas both the first and second phosphorylation step can be taken care of by the VZV-encoded TK. In their 5′-triphosphate form (Figure 3) (as specifically surmised for BVDU), the 5-substituted 2′-deoxynucleosides would then compete with dTTP, thereby acting as an inhibitor/substrate of their target enzyme, the herpes viral DNA polymerase. In their final action, they would engage in hydrogen bonding (i.e., base pairing with adenine).

## 5. Acyclic Nucleoside Analogues

That BVDU as a thymidine analogue would be recognized as a substrate by the viral thymidine kinase (TK) could be anticipated [17], but that acyclovir could be specifically recognized as a substrate by the HSV-encoded TK came as a total surprise [18]. This resulted in the phosphorylation of ACV to its monophosphate (ACV-MP) (Figure 4), which upon further phosphorylation by cellular kinases to its diphosphate (ACV-DP) and triphosphate (ACV-TP), then yielded the active metabolite, capable of base pairing with cytosine (G⋮C) and acting as a competitive inhibitor of dGTP. Since the original description of its antiviral activity against HSV [19], acyclovir has become the “gold standard” for the treatment of both HSV-1 and HSV-2 infections by both the systemic (parenteral) and local (topical) route(s). Its active metabolite was ascertained to be ACV-TP [20]. For oral administration, acyclovir has been replaced by its aminoacyl (i.e., valyl) valaciclovir (Figure 4), which was originally described among a number of aminoacyl esters designed to increase the aqueous solubility of acyclovir [21]. For the treatment of human cytomegalovirus (HCMV) infections, ganciclovir (Figure 4) proved far superior to acyclovir (as originally shown by John C. Martin and Julien P. Verheyden at Syntex), and as demonstrated for valaciclovir, the valyl ester of ganciclovir (valganciclovir) proved superior to ganciclovir itself in the treatment of HCMV infections. Independently from acyclovir and ganciclovir, penciclovir was divulged as an anti-HSV agent that was also active against VZV. To enhance its oral bioavailability, penciclovir was then further converted to famciclovir (Famvir^®^) (Figure 4), which differs from penciclovir by the fact that it is doubly acetylated in the side chain and deoxygenated at the C-6 position of the guanine moiety.

## 6. 2′-,3′-Dideoxynucleoside Analogues

That 2′-,3′-dideoxynucleosides (ddNs) could serve as chain terminators in DNA synthesis was evident from the observations of Sanger et al. [22]. The ddNs, ddAdo, ddGuo, ddCyt and ddThd were shown to be useful for the sequencing of DNA based on the principle that the newly formed DN chain could be terminated at either A, G, C or T with either ddATP, ddGTP, ddCTP or ddTTP respectively, as the substrate, thus revealing the position (sequencing) of A, G, C or T in the DNA chain.

This principle has been applied, albeit inadvertently, in the identification of ddNs, then called nucleoside reverse transcriptase inhibitors (NRTIs), as inhibitors of HIV replication. The NRTI era started with the discovery of AZT (Figure 5A) [23], followed by ddI and ddC [24], d4T [25,26,27], 3TC [28], ABC [29] and, finally, (-)FTC [30] (Figure 5A) (Hivid^®^ has in the meantime been abandoned for clinical use in the treatment of HIV infections because of unacceptable (neuro)toxicity).

All other NRTIs are still available, although they are mostly used in combination with other anti-HIV drugs. To be active as anti-HIV agents, they need to be phosphorylated successively to their 5′-monophosphate (MP), 5′-diphosphate (DP) and 5′-triphosphate (TP) form (Figure 5B) before the latter will act, as a chain terminator, in competition with dTTP, dATP, dCTP or dGTP, respectively (Figure 5B). For AZT, ddC, d4T, 3TC and (-)FTC, this phosphorylation pathway is straightforward. For ddI and ABC, it is interrupted by a conversion of the hypoxanthine to adenine moiety and of abacavir (ABC) to carbovir (CBV), respectively, at the 5′-monophosphate 2′,3′-dideoxynucleoside stage (Figure 5B). As DNA chain terminators, all 5′-triphosphate 2′,3′-dideoxynucleosides engage in hydrogen bonding (base pairing) following the A:T or G⋮C principle.

## 7. ANPs: Acyclic Nucleoside Phosphonates

The discovery of the ANPs was heralded with the advent of (*S*)-9-(3-hydroxy-2-phosphonylmethoxypropyl)adenine (HPMPA) and 9-(2-phosphonyl)adenine (PMEA) [31]. PMEA was eventually commercialized as the oral prodrug adefovir dipivoxil (Hepsera^®^) for the treatment of HBV (hepatitis B virus) infections. Within one year after HPMPA and PMEA, the cytosine counterpart of HPMPA, (*S*)-1-(3-hydroxy-2-phosphonylmethoxypropyl)cytosine (HPMPC), was recognized as a broad-spectrum anti-DNA virus agent [32]. Nine years later, it would be commercialized as Vistide^®^ (cidofovir) for the treatment of human cytomegalovirus (HCMV) retinitis in AIDS patients. In 1993, (*R*)-9-(2-phosphonylmethoxypropyl)adenine (PMPA) would be announced as a potent inhibitor of HIV (human immunodeficiency virus) replication [33], and the oral prodrug of PMPA, in the meantime named tenofovir, would be commercialized in 2001 as Viread^®^ (tenofovir disoproxil fumarate, TDF). All three ANPs, adefovir, tenofovir and cidofovir, act in a similar fashion, that is, by base pairing through the formation of hydrogen bonds: adefovir and tenofovir with thymine and cidofovir with guanine (Figure 6).

What has remained mysterious is that the ANPs adefovir, tenofovir and cidofovir are restricted in their activity spectrum to DNA viruses, i.e., HBV, CMV (and other herpesviruses) and retroviruses (such as HIV), which, while being RNA viruses, behave as DNA viruses after their RNA genome has been transcribed to DNA by their reverse transcriptase (RT).

## 8. O-DAPYs

The acyclic 2,4-diaminopyrimidine nucleoside phosphonates (PMEO and PMPO DAPY derivatives) were identified in early 2002 [34,35]. The PMEO and PMPO pyrimidine ANP derivatives exhibit pronounced activity against both retroviruses (i.e., HIV) and hepadnaviruses (i.e., HBV) [36,37]. Further investigations with this class of compounds were hampered by the (premature) death of its pioneer, Antonín Holý, on 16 July 2012. Yet, the fascinating findings obtained for this class of compounds, those being pyrimidine analogues that behave as purine derivatives (Figure 7) [38], justify their further exploration.

The heterocyclic moiety of the O-DAPYs could be considered an uncompleted purine base, thus allowing hydrogen bonding as observed with adefovir and tenofovir. As a consequence, the O-DAPYs should be further pursued for their therapeutic potential in the treatment of those viruses (i.e., HIV, HBV) infections that fall within the realm of adefovir and tenofovir.

## 9. BCNAs (Bicyclic Nucleoside Analogues)

As a prototype of the BCNAs, Cf1743 and its 5′-valyl ester FV-100 (FV being Fermavir) (Figure 8A) still stand out as the most potent antiviral drugs ever described [39]. Cf1743 and FV-100 are exclusively active against VZV (varicella-zoster virus). They exhibit no activity against any other virus.

Among a series of several antiviral agents, including acyclovir and brivudine (BVDU), Cf1743 emerged as the most potent anti-VZV compound. It is evident that the compound needs to be phosphorylated by the virus-encoded thymidine kinase (TK) since it is not active against TK-deficient VZV mutants, but its active metabolite, securing its anti-VZV activity has not been identified. It is assumed that Cf1743, in its 5′-triphosphorylated form, inhibits the VZV DNA synthesis, but how it would do so has never been ascertained.

The hypothesis that CF1743 could be hydrogen-bonded to adenine according to the Watson–Crick type of base pairing does not seem feasible (Figure 8B); thus remains the possibility that Cf1743 may engage in hydrogen bonding (base pairing) with guanine. According to the Watson–Crick rule of base pairing, two instead of three hydrogen bonds may be involved in this type of base pairing of Cf1743.

## 10. Remdesivir

Remdesivir (GS-5734) is a prodrug of a C-nucleoside (Figure 9A) with a purine moiety reminiscent of adenine, which explains why it is able to base-pair with uracil (Figure 9B). After COVID-19 emerged, remdesivir has been granted special approval (as Veklury^®^) for the treatment of SARS-CoV-2 infections [40].

The C-nucleoside part of remdesivir originally featured among a series of 1′-substituted 4-aza-7,9-dideazaadenosine C-nucleosides [41], which showed a broad range of antiviral effects, particularly against HCV (hepatitis C virus). The antiviral activity of this class of compounds was further documented by Mackman et al. [42].

Remdesivir (GS-5734) contains a conspicuous cyano (CN) moiety, which was considered crucial for its antiviral activity [42]. How this cyano group operates has not been ascertained. As a possible hypothesis, it can be proposed that it is further hydrated to a carboxamide (-C≡N + H_2_O → -CO-NH_2_).

## 11. RNA Virus Mutagen

The term RNA virus mutagen was first used by Crotty et al. [43] to indicate the antiviral activity of ribavirin. Ribavirin’s antiviral activity would be exerted directly through lethal mutagenesis of the viral genetic material, leading to RNA virus error catastrophe [44]. In its “anti” conformation, ribavirin, similar to guanine, would base-pair with cytosine (Figure 10A). However, after rotation of the carboxamide moiety, it may also form hydrogen bonds with uracil (Figure 10B) [43]. This would ultimately lead to a transition mutation of G⋮C → Rib:C → Rib:U → A:U.

Ribavirin was first described by Sidwell et al. [45] as a broad-spectrum antiviral agent effective against both DNA and RNA viruses. Its mechanism of antiviral action was attributed to the inhibition of IMP dehydrogenase (IMPDH) [46], thus resulting in the inhibition of the synthesis of GMP (*via* XMP), GDP and GTP (the immediate precursor for the synthesis of (viral) RNA). That ribavirin (and other IMPDH inhibitors) may not exert their antiviral activity against Yellow Fever Virus (YFV) or other flaviviruses and paramyxoviruses by error-prone replication, leading to error catastrophe, has been ascertained repeatedly [47,48].

## 12. Favipiravir

Favipiravir is a pyrazine analogue (Figure 11A) equipped with a carboxamide moiety similar to that present in ribavirin. Akin to orotic acid, favipiravir can be converted through the help of PRPP (phosphoribosylpyrophosphate) transferase to a phosphoribosyl derivative (which can then be further phosphorylated to its 5-di- and 5-triphosphate).

Favipiravir has proven efficacious in the treatment of influenza [49] and various hemorrhagic fever (HF) virus infections [50].

Through the carboxamide moiety, favipiravir can engage in base pairing with both cytosine and uracil (Figure 11B), similar to ribavirin. 

Favipiravir leads to facile insertion into viral RNA, provoking C-to-U and G-to-A transitions, resulting in SARS-CoV-2 lethal mutagenesis [51,52]. On influenza viral polymerase, it acts as a unique delayed chain terminator [53]. Favipiravir and molnupiravir exert their antiviral action through lethal mutagenesis, which may limit their extended use in antiviral therapy because of carcinogenic risks and genotoxicity [54].

## 13. Molnupiravir [N^4^-Hydroxycytidine (NHC)]

Against SARS-CoV-2, molnupiravir (Figure 12A) may be 100 times more potent as an antiviral agent than ribavirin or favipiravir [55]. It prevents the replication of SARS-CoV-2 by fostering error accumulation in a process referred to as “error catastrophe” [56]. NHC can form stable base pairs with either guanine or adenine (Figure 12B) in the RdRp active center so that the polymerase escapes proofreading and synthesizes mutated RNA. This mutagenesis mechanism probably extends to various viral RNA polymerases pertaining to the broad-spectrum antiviral activity of molnupiravir [57]. That molnupiravir is leading to error catastrophe is not surprising since Janion and Glickman [58] have already revealed the potential of N^4^-hydroxycytidine to cause AT → GC transitions.

## 14. Sofosbuvir

As a uracil derivative, sofosbuvir is faithful in its base pairing with adenine (Figure 13).

This base pairing underlies the mechanism of action of sofosbuvir at the HCV RdRp level. The structural basis for RNA replication by the HCV polymerase has been extensively described [59]. Sofosbuvir has been hailed as a novel treatment option for chronic HCV infection [60,61,62]

## 15. Addendum: Resistance to Glycopeptide Antibiotics (i.e., Vancomycin)

A unique example of the role of hydrogen bonding can be found in the mechanism of resistance development to certain antibiotics, i.e., glycopeptides.

Prototypes of glycopeptide antibiotics are vancomycin (Figure 14A1) and teicoplanin (Figure 14A2).

Resistance development towards these antibiotics (Figure 14B) requires the production of a variety of enzymes, i.e.:Van H reductase, which metabolizes pyruvate to D-lactate;Van X dipeptidase, which cleaves D-Ala-D-Ala into 2 D-Ala;Van A ligase, which joins D-Ala onto D-lactate.

D-Ala-D-lactate is then incorporated instead of D-Ala-D-Ala during NAM-decapeptide formation, an essential component in building the peptidoglycan part of the bacterial cell wall. The Van Y carboxypeptidase leads to the formation of NAM-nonapeptide with only one (instead of two) D-Ala residues, thus resulting in the formation of a defective peptidoglycan.

In the biosynthesis of vancomycin-sensitive bacteria, D-Ala-D-Ala is needed to permit hydrogen bonding with vancomycin (Figure 14C1). When D-Ala-D-Ala is replaced by D-Ala-D-Lactate, no such hydrogen bonding is possible (Figure 14C2), resulting in the emergence of resistance of the bacteria to vancomycin.

## 16. Conclusions

Here, I have described how several antiviral agents owe their activity to hydrogen bonding (base pairing), for example, acyclic nucleoside phosphonates (ANPs): adefovir, tenofovir, cidofovir and O-DAPYs. This list could be further extended to sofosbuvir and remdesivir, whereas for others, such as Cf1743, this hydrogen bonding still remains to be established. For several antiviral drugs such as acyclovir, ganciclovir and penciclovir, it is well known that as guanine derivatives, they base-pair through hydrogen bonding with cytosine to inhibit the viral DNA polymerase. The uracil derivatives IDU, TFT and BVDU do so by base pairing with adenine. In the present review article, the importance of hydrogen bonding (base pairing) has also been emphasized in the antiviral action of ribavirin, favipiravir and molnupiravir. In the latter instance, base pairing inevitably leads to lethal mutagenesis (error catastrophe), which, on the one hand, may explain the observed antiviral effects, i.e., against COVID-19, but, on the other hand, may account for undesirable mutagenic effects on the host organism. As an example of hydrogen bonding not related to A:T or G⋮C base pairing, hydrogen bonding, or the lack thereof, has been implicated in a specific case of antibiotic resistance. When developing resistance to glycopeptide antibiotics such as vancomycin, bacteria may have acquired the possibility of incorporating D-Ala-D-lactic acid, instead of D-Ala-D-Ala, into the peptidoglycan of their cell wall, thus preventing the hydrogen bonding normally required for bacterial sensitivity towards the antibiotic.

## Figures and Tables

**Figure 1 viruses-15-01145-f001:**
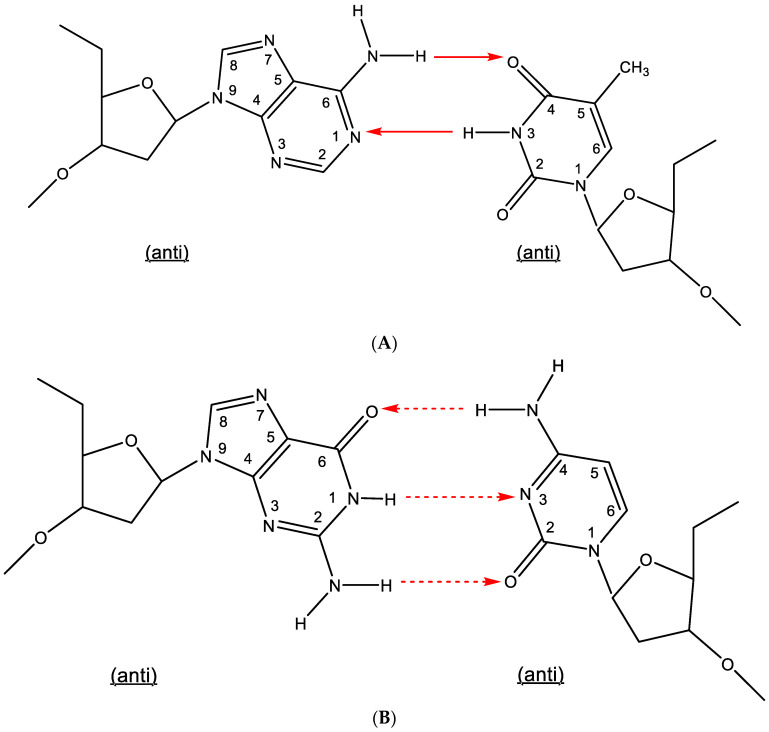
(**A**). Watson–Crick type of base pairing (adenine-thymine). (**B**). Watson–Crick type of base pairing (guanine-cytosine). The red arrows indicate the direction of the hydrogen bonding.

**Figure 2 viruses-15-01145-f002:**
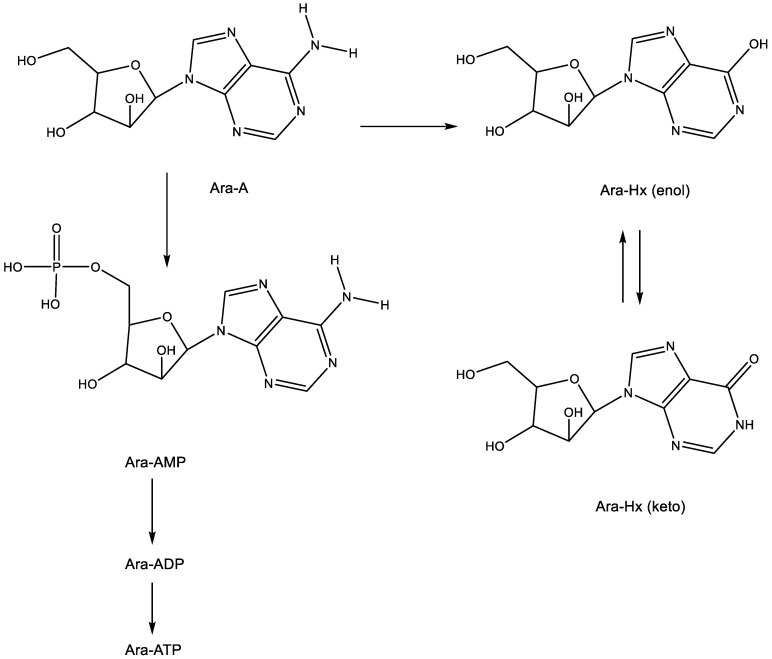
Metabolism and degradation of Ara-A. The arrows indicate the metabolic conversions.

**Figure 3 viruses-15-01145-f003:**
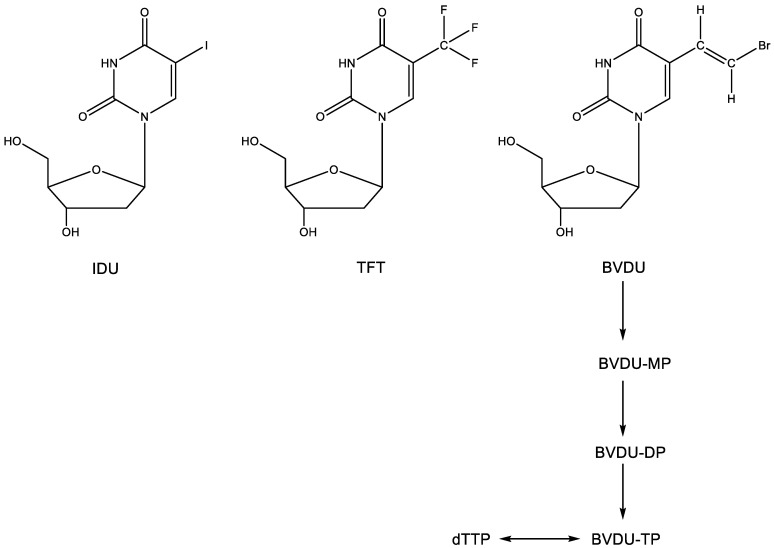
2′-Deoxyuridines IDU, TFT and BVDU.

**Figure 4 viruses-15-01145-f004:**
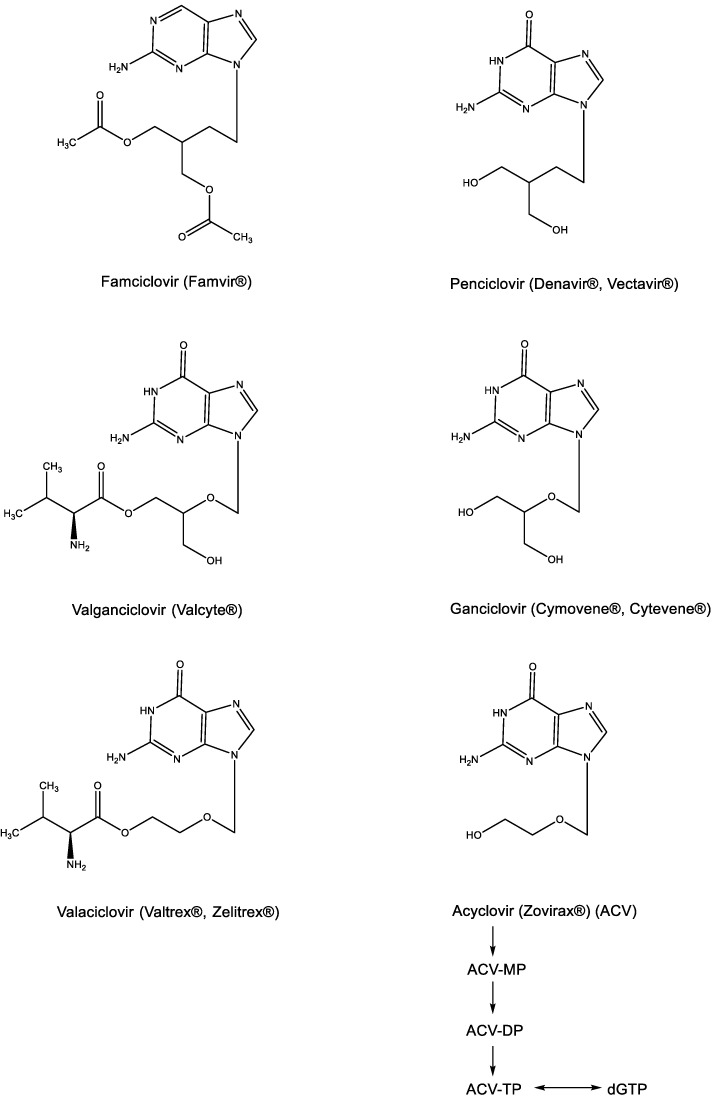
Acyclic nucleoside analogues: Acyclovir, Ganciclovir, Penciclovir and their prodrugs (Valaciclovir, Valganciclovir and Famciclovir).

**Figure 5 viruses-15-01145-f005:**
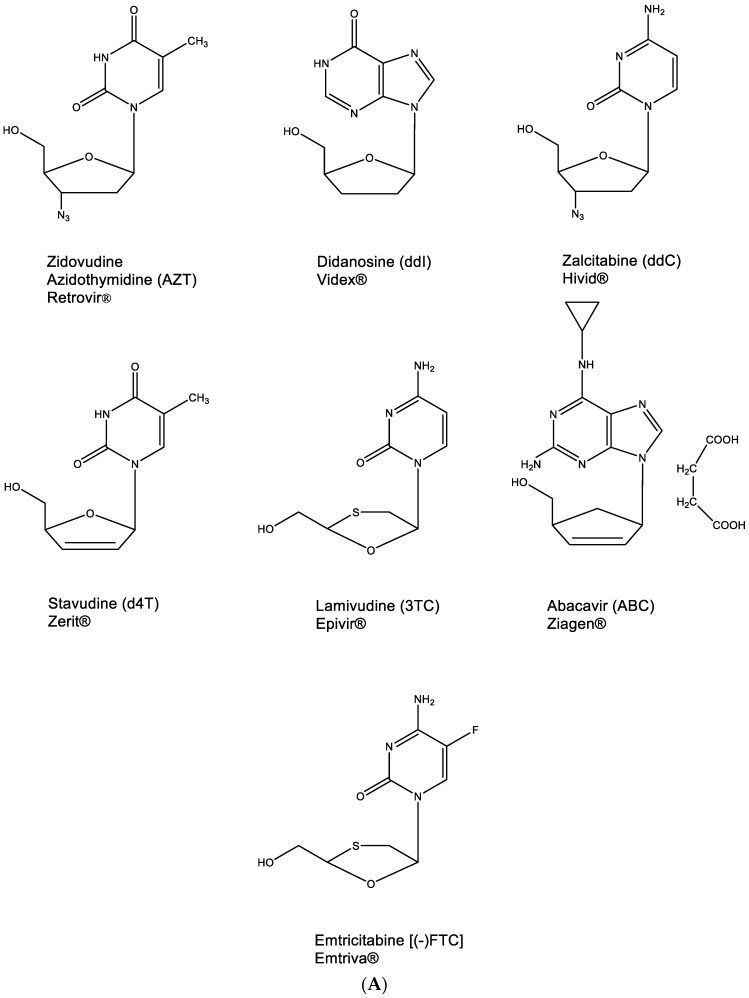
(**A**). 2′,3′-Dideoxynucleoside analogues (ddNs). (**B**). Metabolism of ddNs.

**Figure 6 viruses-15-01145-f006:**
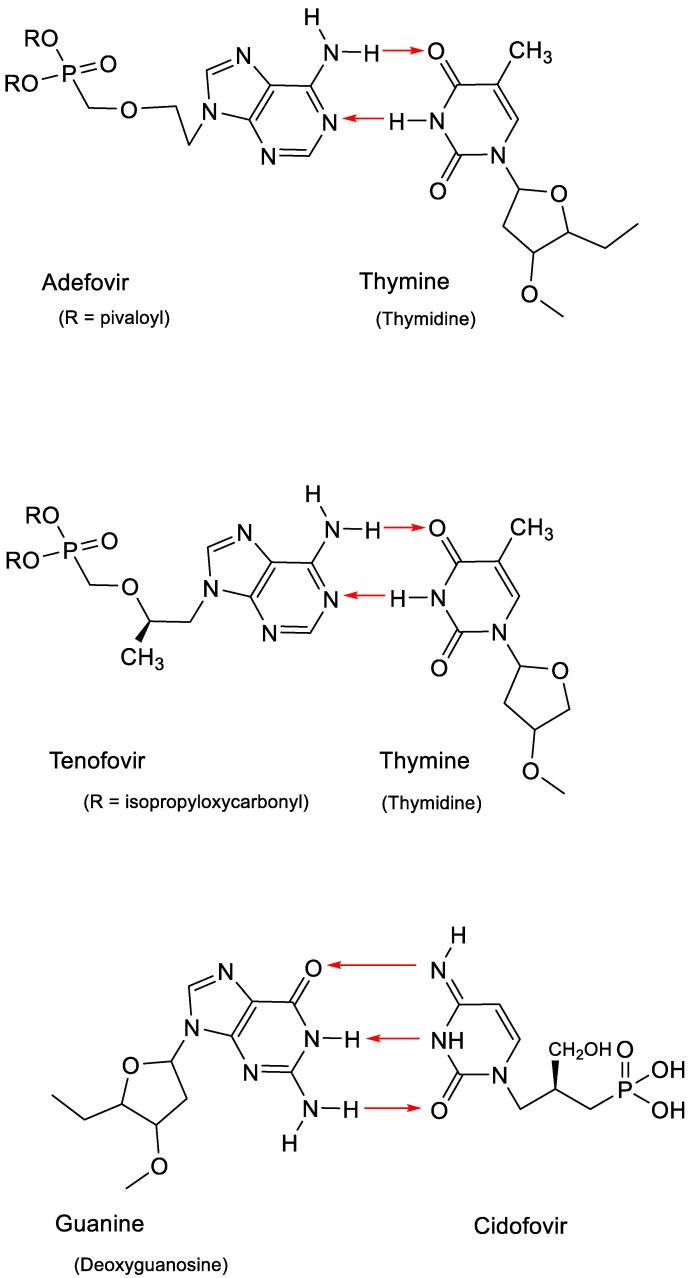
Base pairing of thymine with adenine in adefovir or tenofovir and guanine with cytosine in cidofovir.

**Figure 7 viruses-15-01145-f007:**
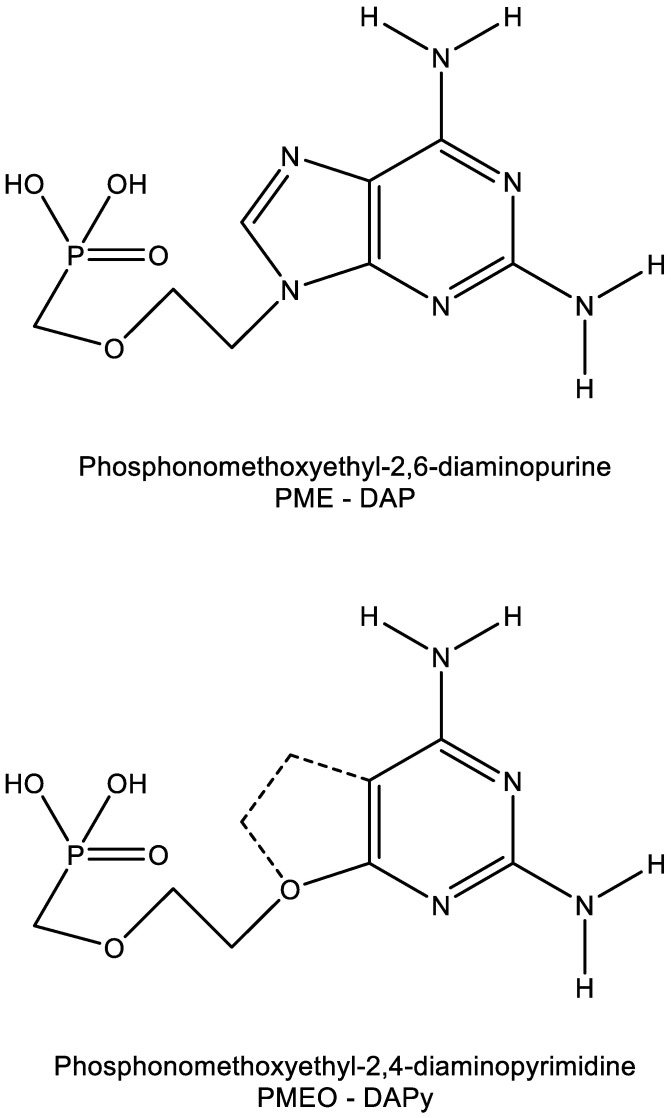
Similarity in molecular structure between PME-DAP (P = purine) and PMEO-DAPy (Py = pyrimidine).

**Figure 8 viruses-15-01145-f008:**
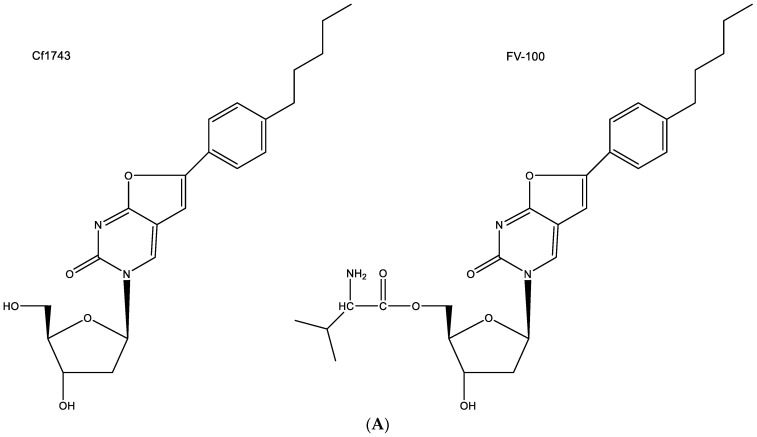
(**A**). structures of Cf1743 and FV-100. (**B**). CF1743 cannot base-pair with adenine according to Watson–Crick type of base pairing (only one hydrogen bond possible). The red arrows indicate the direction of the hydrogen bonding.

**Figure 9 viruses-15-01145-f009:**
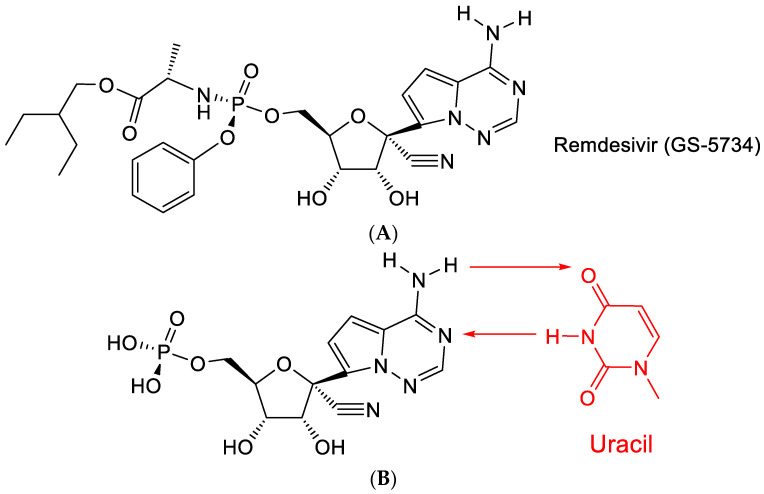
(**A**). Structure of Remdesivir. (**B**): Remdesivir: base pairing with uracil. The red arrows indicate the direction of the hydrogen bonding.

**Figure 10 viruses-15-01145-f010:**
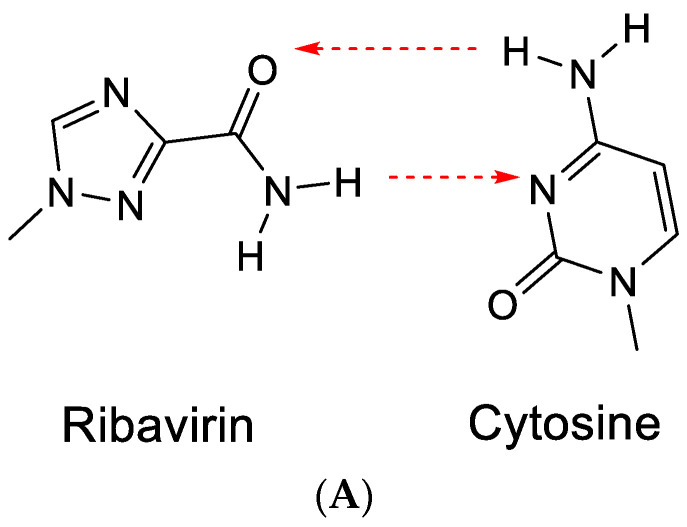
(**A**). Ribavirin can base-pair with cytosine. (**B**). Ribavirin can base-pair with uracil. The red arrows indicate the direction of the hydrogen bonding.

**Figure 11 viruses-15-01145-f011:**
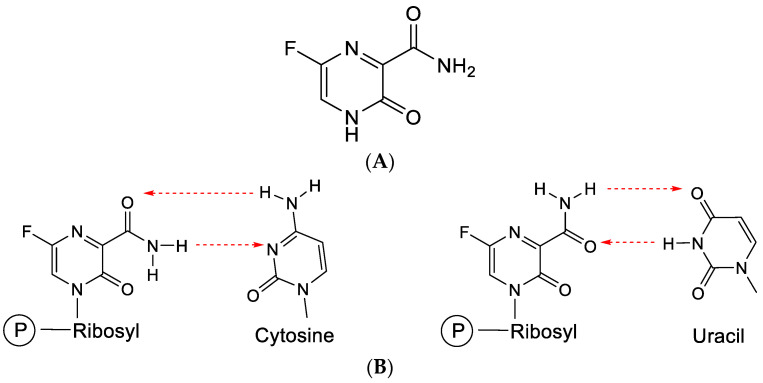
(**A**). Structure of favipiravir. (**B**). Favipiravir (following conversion to its ribonucleotide) can base-pair with cytosine or uracil. The red arrows indicate the direction of the hydrogen bonding.

**Figure 12 viruses-15-01145-f012:**
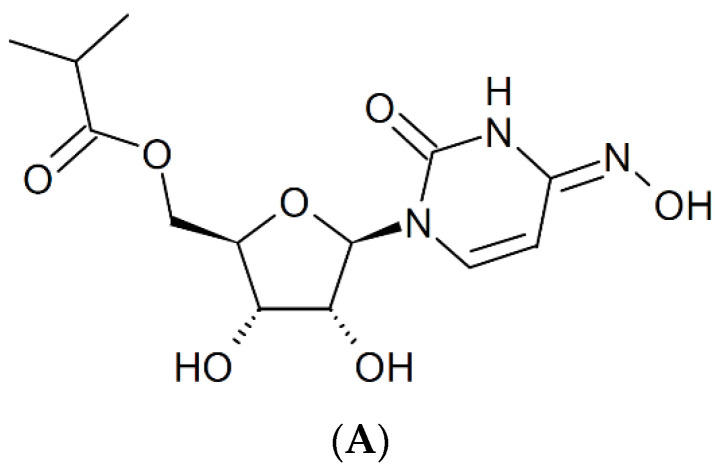
(**A**). Structure of molnupiravir. (**B**). Base pairing of molnupiravir. The red arrows indicate the direction of the hydrogen bonding.

**Figure 13 viruses-15-01145-f013:**
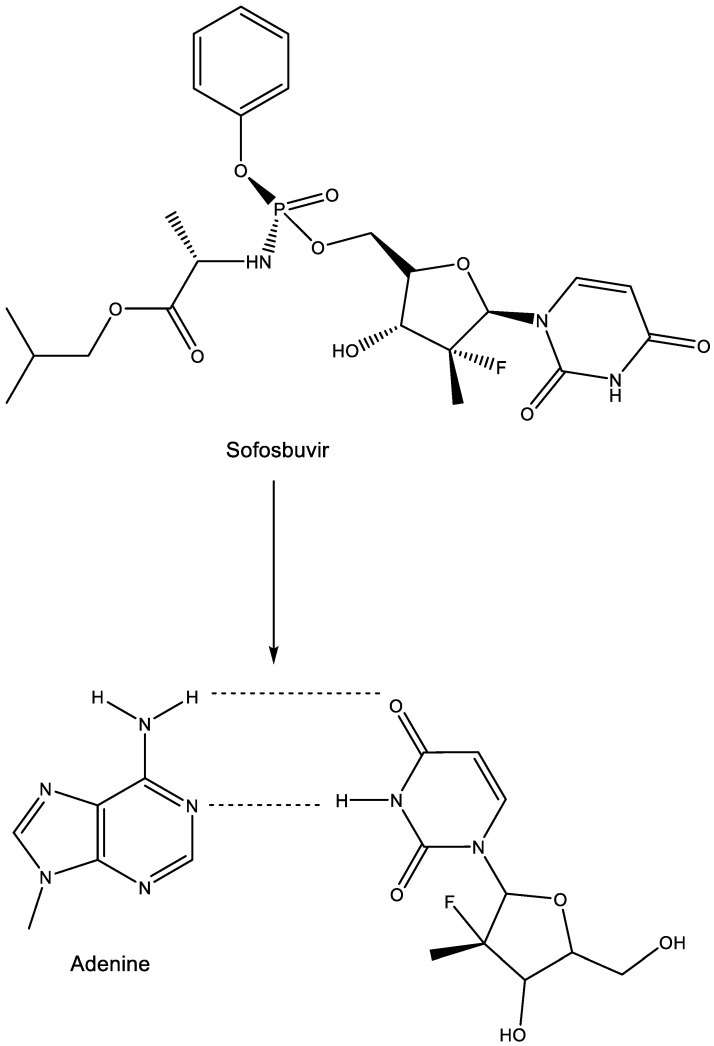
Sofosbuvir: prodrug of metabolite base pairing with adenine.

**Figure 14 viruses-15-01145-f014:**
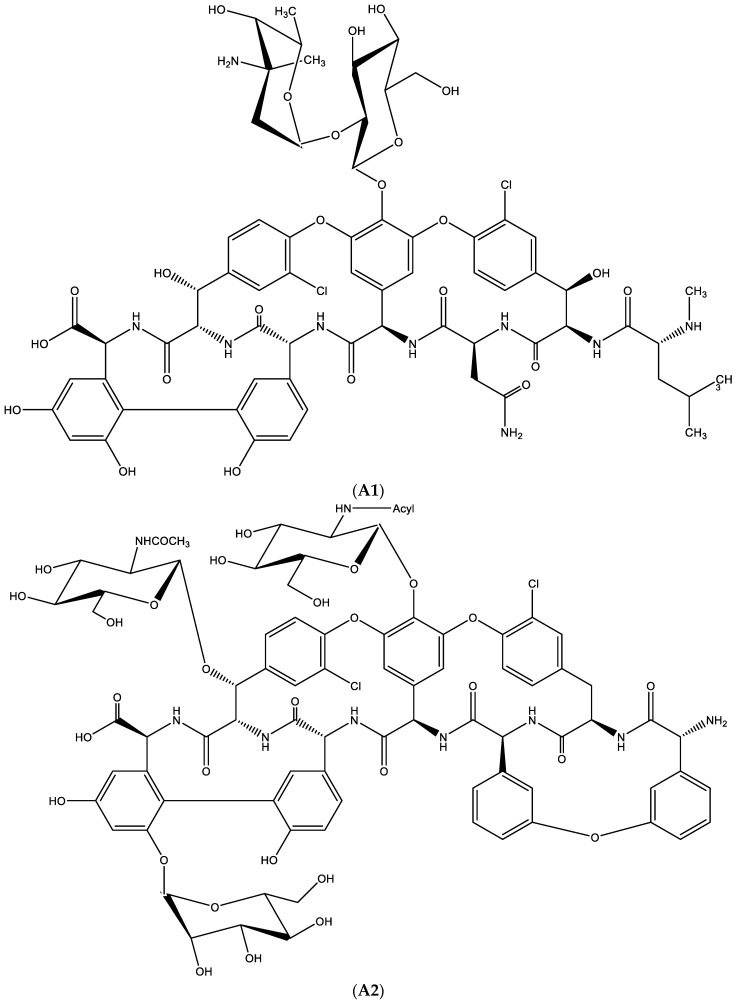
(**A1**). Glycopeptide antibiotic: vancomycin. (**A2**). Glycopeptide antibiotic: teicoplanin. (**B**). Resistance towards vancomycin. (**C1**). D-Ala-D-Ala is needed to permit hydrogen bonding with vancomycin. (**C2**). When D-Ala-D-Ala is replaced by-D-Ala-D-Lactate, no such hydrogen bonding is possible.

## Data Availability

Not applicable.

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
