# Peer review of "Hydrogen Bonding (Base Pairing) in Antiviral Activity"

_viruses, 2023, doi:10.3390/v15051145_

Round 1

Reviewer 1 Report

Title: Hydrogen bonding in antiviral activity.

The manuscript reviews that hydrogen bonding is crucial in the activity and the modification of antiviral agents using the detailed examples. To increase readability, modifications are need in this manuscript. 

1. This manuscript include the contents on hydrogen bonds related to solubility, bioavailability as well as antiviral activity of antiviral agents. However, they are not mentioned in Abstract and Introduction. 

2. The designation of hydrogen bonds (using colors) in all figures are helpful to understand Results and figures. 

3. The title of Fig 2 is missing. 

4. The examples on the hydrogen bonds between antiviral agents and viruses in figures. There are not the hydrogen bonds but only the structure of antiviral agents in many figures. Readers might be interested in this point. 

5. If it is possible, the categorizing antiviral agents is needed depending on the roles of hydrogen bonds. 

6. The author’s conclusion about the correlation between hydrogen bonding and broad spectrum of antiviral effects.

Author Response

  1. This is correct, but are  thoroughly explained in the rest of the manuscript.
  2. OK
  3. This has been added
  4. Hydrogen bonds between antivirals in viruses were not envisaged in this article
  5. This is not possible
  6. This was not envisaged in this article

Reviewer 2 Report

The review provides a very interesting and profound discussion of the antivirals acting throught the hydrogen bonding. The idea of the paper is very interesting for the audience of virologists. However, the paper needs significant improvement, i.e. both in terms of editing and the content.

In particular, the paper:

- contains a lot of small editing errors, such as page 99 (Capital letter missing), the phrases cannot be started from That... etc.

- Some figures' legends are missing  (Figure 5b)

- Chaper 13 is missing its rationale - why is it here? The paper is about antivirals, and not antibiotics.

- I guess that some more pharmacological data are necessary (toxicity, affective doses, etc) - to help the readers understand why some of these drugs were dropped out, while some other proved to be working well. For some antiviral such information is provided, while for some others - not.

- Perhaps a kind of a table summarising which drug has been studied for which virus and how far used in clinics (clinical trials) whould be useful.

Author Response

  1. Line 99: correction to capital letter
  2. Missing legend of Figure 5b has been added
  3. Chapter 13  has now been moved to the end (section 14) as Addendum. Its rationale has been added
  4. More pharmacological data: was beyond the scope of the present article
  5. Clinical trials fall again beyond the scope of the present article

Reviewer 3 Report

In this manuscript, the author summarizes a panel of antiviral drugs that recognize their viral targets by hydrogen bonding, and discusses their mechanisms of action. It primarily focuses on the nucleoside analogue compounds that mimic nucleotide substrates to interfere with DNA/RNA synthesis during virus replication. These compounds have long been thought of as good candidates for developing broad-spectrum antiviral drugs, and a few of them have been successfully applied to the treatment of some viral infections. On the other hand, due to the similarity of these molecules to nucleotide substrates, they may also be utilized by cellular polymerases to result in side effects. Therefore, modification/optimization of these compounds is an important topic in this field for better discriminating viral polymerases over cellular polymerases. Overall, this manuscript provides a comprehensive overview of many available nucleotide analog drugs/compounds, which is of general interests to biologists in different fields. However, some aspects regarding the mechanism of action for a few such molecules are not adequately discussed, and some aspects are not accurately described, which should be substantially revised before acceptance for publication.

1.      This manuscript mainly focuses on nucleotide analog compounds that execute their functions by base-pairing (hydrogen-bonding) with template DNA/RNA. This background is explicitly laid out in the introduction section, and fits the logic of the review. I would suggest to replace the title “Hydrogen bonding …” with “Base pairing …”. Also the section about glycopeptide antibiotics is better to be removed. If the author wants to otherwise discuss “Hydrogen bonding”, then much more molecules should be included in addition to nucleotide analogs.

2.      Lines 221-223, the hypothetical mechanism of action of Remdesivir is inaccurate according to the available structural studies on SARS-CoV-2 polymerase that incorporate Remdesivir into RNA product, which have also provided some insights into the function of the cyano group. The base pairing model in Figure 9 is not supported by the available structures. Remdesivir is actually an Adenosine analog that base pairs with U/T.

3.      Lines 256-257, “Whether favipiravir leads to error- prone replication or error catastrophe, has not been ascertained.” Actually, some studies have shown, at least at cellular level, that Favipiravir treatment can lead to accumulation of mutations in the genome of SARS-CoV-2 at a higher rate (Shannon et al., 2020). A few studies have reported the structural basis of Favipiravir incorporation into RNA product by SARS-CoV-2 polymerase. Different from Remdesivir that leads to aberrant termination in RNA synthesis, Favipiravir will not directly terminate RNA synthesis but instead result in mutations by non-unique base pairing with C and U. These studies should be included and discussed in detail to reveal the current status of our understanding on the mechanisms of these drugs.

4.      Similar to the previous point, the structures of Ribavirin and penciclovir being incorporated into RNA product by SARS-CoV-2 polymerase have been reported (https://www.biorxiv.org/content/10.1101/2020.11.01.363812v1). The underlying mechanism of action should be discussed in more detail, and the base pairing model should be revised according to the available structures, instead of using hypothetical models.

5.      The discussion about sofosbuvir is too brief and cannot reveal the mechanism of its antiviral effect. Based on the structural and biochemical studies showing how sofosbuvir is recognized and incorporated into RNA product by HCV polymerase (Appleby, T. C. et al. 2015), the author should provide more detailed discussions and explain how it inhibits viral replication.

Author Response

  1. The term "base pairing" has been added to the title. The section on glycopeptide antibiotics has been moved to the end as Addendum
  2. This has already been shown in Fig. 9b. Further speculation on possible hydrogen bonding with cytosine (Fig. 9c) has been deleted.
  3. References to Shannon et al (ref. 51) and additional references (52, 53, 54) have been added
  4. This reference has not been published as a regular paper and was deleted
  5. Little is known about the mode of action of sofosbuvir. Therefore, the study of Appleby et al (ref. 59) is highly welcome

Reviewer 4 Report

The review should include some additional discussion and references where the antiviral action of ribavirin is associated with lethal mutagenesis = error catastrophe, in the explanatory context of the role of hydrogen bonding base pairing. The references #47-48 provide alternative mechanisms of activity of ribavirin that are independent of error-prone replication, so there should be some citations of viruses where lethal mutagenesis/error catastrophe occur.

Check the text to ensure that identification of the names of commercialized agents are defined appropriately to avoid possible confusion. For example, on Page 6, line 124 the drug name Hivid® appears without having been previously defined.

The structures of the sugars in the T & G nucleotide units that are shown hydrogen bonding in Figure 6 are incomplete at the 3' and 5' positions and should be corrected to a standardized representation.

Author Response

  1. Citations of references on lethal mutagenesis are amply provided in references 47 and 48. That the predominant mechanism by which ribavirin inhibits the replication of four flavi- and two paramyxoviruses (references 47 and 48) is not error catastrophe but inhibition of cellular IMP dehydrogenase (IMPDH activity and thus depletion of intracellular GTP pools, has again been emphasized in the present perspective article
  2. Hivid© has been shown in Figure 5a
  3. In Fig. 6 a mistake was indeed found in the cytosine paired with guanine, and this has been corrected